# DockGame: Cooperative Games for Multimeric Rigid Protein Docking

## Abstract

Protein interactions and assembly formation are fundamental to most biological processes. Predicting the assembly structure from constituent proteins – referred to as the protein docking task – is thus a crucial step in protein design applications. Most traditional and deep learning methods for docking have focused mainly on binary docking, following either a search-based, regression-based, or generative modeling paradigm. In this paper, we focus on the less-studied *multimeric* (i.e., two or more proteins) docking problem. We introduce DockGame, a novel game-theoretic framework for docking – we view protein docking as a *cooperative game* between proteins, where the final assembly structure(s) constitute stable *equilibria* w.r.t. the underlying game potential. Since we do not have access to the true potential, we consider two approaches - i) learning a surrogate game potential guided by physics-based energy functions and computing equilibria by simultaneous gradient updates, and ii) sampling from the Gibbs distribution of the true potential by learning a diffusion generative model over the action spaces (rotations and translations) of all proteins. Empirically, on the Docking Benchmark 5.5 (DB5.5) dataset, DockGame has much faster runtimes than traditional docking methods, can generate *multiple* plausible assembly structures, and achieves comparable performance to existing binary docking baselines, despite solving the harder task of coordinating multiple protein chains.

## 1 Introduction

Protein function is often dictated by interactions with other proteins, forming assemblies that regulate most biological processes. Predicting these assembly structures from constituent proteins – referred to as the (multimeric) protein docking task – is crucial in protein engineering applications. Traditional and deep learning methods for protein docking have largely focused on binary docking (i.e., two proteins). These methods either use a scoring function to rank millions of candidate assemblies, or model assembly structure prediction as a regression (Ganea et al., 2022) or generative modeling problem (Ketata et al., 2023) over the space of rotations and translations.

Only a handful of approaches exist for the harder *multimeric* docking task (i.e. involving more than two proteins). A key challenge here is that the space of possible actions – rotational, translational and *optionally* conformational changes, grows combinatorially with the number of proteins. Traditional methods for this task therefore first generate pairwise docking candidates and then combine them using combinatorial optimization algorithms, making them very inefficient and slow. Deep learning approaches for multimeric docking, while faster, still utilize some aspect of pairwise decomposition in their architectures – either in generating multiple sequence alignments (Evans et al., 2021), or pairwise roto-translation predictions (Ji et al., 2023). Moreover, they are both limited to predicting a single assembly structure[1] (while in fact there might be multiple plausible structures).

In this work, we introduce DockGame, a novel game-theoretic framework for the rigid multimeric docking problem: we view docking as a game between protein chains or their relevant subassemblies (smaller assemblies made up of chains), where the final assembly structure(s) constitute stable *equilibria*. To the best of our knowledge, this is the first approach connecting game theory with protein docking. In particular, we model docking as a *cooperative* game, where all proteins have

---

[1]AlphaFold-Multimer uses an ensemble of 5 models, each with a single assembly structure prediction.

Figure 1: Overview of DOCKGAME. We introduce a game-theoretic framework for rigid multimeric docking: we view assembly structures as equilibria w.r.t to the potential of an underlying cooperative game. In practice, however, we do not have access to such potential function. We propose two approaches to tackle this. *Left*: We learn a surrogate potential using supervision from traditional physics-based scoring functions, and compute equilibria via a gradient-based scheme. *Right*: Viewing assembly structures as samples from the Gibbs distribution of the underlying potential, we learn a diffusion generative model over the roto-translation action spaces of all game agents.

aligned interests (e.g., improved assembly energetics) described through an underlying potential, and the action space for each protein corresponds to the continuous space of rotations and translations. Intuitively, this allows us to simultaneously model interactions between all proteins, and decouple the combinatorial action space into individual ones.

In practice however, we do not have access to the true underlying potential. To tackle this problem, we propose two approaches which we summarize in Figure 1. Our first approach employs *supervised learning*, learning a differentiable analogue of traditional (and black-box) physics-based scoring functions. This allows us to compute equilibria via a gradient-based scheme over the space of roto-translations w.r.t the learnt potential. While simple, efficient, and generalizable to various objectives, it crucially relies on the supervision signal, which may not accurately reflect the game structure underlying observed assemblies. Our second approach, instead is *self-supervised*, learning solely from assembly structures in the data by interpreting them as samples from the Gibbs distribution of the underlying potential. To this end, we then define a diffusion generative model (DGM) over the joint roto-translation spaces of all players (relative to a fixed player), which can then be trained with standard denoising score-matching objectives.

To summarize, we make the following contributions:

- We formulate (rigid) multimeric protein docking as a cooperative game between proteins, where the final assembly structure(s) constitute equilibria w.r.t a underlying potential. To the best of our knowledge, this is the first work connecting game theory to protein docking. The game-theoretic framework also allows us to compute multiple equilibria.

- To compute plausible equilibria, we propose two different approaches. Our first approach utilizes supervision from scoring functions, e.g. physics-based, to learn a surrogate game potential and computes equilibria via gradient-based learning updates. Our second approach, instead, is self-supervised in that only uses observed assemblies data to train a DGM that can efficiently sample from the game equilibria distribution.

- We evaluate DOCKGAME on the Docking Benchmark 5.5 (DB5.5) dataset. While DB5.5 has been traditionally used for binary docking evaluation, many proteins in the dataset contain multiple chains, allowing us to evaluate multimeric docking. Despite the harder task, DOCKGAME generates multiple plausible assemblies, and achieves comparable (or better) performance on the commonly used Complex-RMSD (C-RMSD) and TM-score metrics, while exhibiting orders of magnitude faster runtimes than traditional docking methods.

## 2 RELATED WORK

**Protein Docking**  Traditional protein docking methods (Chen et al., 2003; Mashiach et al., 2010; Kozakov et al., 2017; de Vries et al., 2015; Yan et al., 2020) have largely focused on the binary docking setting. These methods typically generate a large number (possibly millions) of candidate complexes, which are then ranked using a scoring function. Finally, top-ranked candidates can undergo further refinement using other energetic models. Deep learning approaches for protein docking have also mainly focused on the rigid binary docking setting, learning the best roto-translation of one protein (ligand) relative to the other protein (receptor) (Ganea et al., 2022), or a distribution thereof (Ketata et al., 2023). Different from all these works, we focus on the less explored (rigid) multimeric docking setting, where the goal is to the predict the assembly structure of $\geq 2$ proteins.

A key challenge in the multimeric docking task, is to efficiently explore the combinatorial space of possible relative motions between proteins. Traditional approaches, such as MULTI-LZERD (Esquivel-Rodríguez et al., 2012), achieve this by generating pairwise docking candidates, which are then combined with combinatorial assembly algorithms, with a similar algorithm in more recent works (Bryant et al., 2022). The only deep learning approaches for multimeric docking, ALPHAFOLD-MULTIMER (Evans et al., 2021), and SYNDOCK (Ji et al., 2023), also adopt pairwise decompositions, either in generating paired multiple sequence alignments, or in predicting pairwise roto-translations. In contrast, we propose a novel game-theoretic framework for multimeric docking based on cooperative games, and compute assembly structures (equilibria) efficiently using gradient-based schemes that decouple the search space over individual proteins. Additionally, game-theory offers a natural way to compute *multiple* equilibria, while previous multimer docking methods are limited to predicting a single assembly structure.

**Diffusion Generative Models for Proteins and Molecules**  Diffusion generative models have becoming increasingly popular in the molecular modeling community, used in tasks like conformer generation (Xu et al., 2022; Jing et al., 2022), molecular and protein docking (Qiao et al., 2022; Corso et al., 2023; Ketata et al., 2023), protein backbone generation (Watson et al., 2022; Yim et al., 2023), and generative structure prediction (Jing et al., 2023). The closest related work to ours are (Corso et al., 2023; Ketata et al., 2023), of which our diffusion generative model for the cooperative game can be interpreted as a multi-agent extension. To our best knowledge, this is the first approach of a "multi-agent" diffusion generative model in the context of multimeric docking (and cooperative games). In this regard, our score network is parameterized such that it can handle a varying number of proteins during training and inference – a fact of potentially more general interest.

## 3 PRELIMINARIES AND BACKGROUND

In this section, we introduce relevant preliminaries and background that form the basis for the remainder of the paper. Throughout, we use the term protein to denote both individual protein chains, and any subassemblies made up of protein chains. We also use $[N]$ to denote the set $\{1, 2, \ldots, N\}$.

**Problem Definition.**  In the multimeric protein docking task, the goal is to predict the protein assembly structure, given 3D structures of the constituent protein chains. The 3D structure of each protein chain $i \in [N]$ is described by a point cloud $\mathbf{X}_i \in \mathbb{R}^{n_i \times 3}$ where $n_i$ is the number of chain's residues, and the residue position corresponds to the coordinates of its $C\alpha$ atom.

**Game-Theoretic Concepts.**  Game theory provides a natural framework to model and analyze systems composed of multiple self-interested agents. Formally, a game can be described by $N$ agents, indexed $i \in [N]$. Each agent $i$ is equipped with an action set $\mathcal{A}_i$, and a cost function $f_i : \prod_{i=1}^{N} \mathcal{A}_i \to \mathbb{R}$, that maps the actions chosen by all agents to a scalar cost for agent $i$. Solution concepts to games are defined by *equilibria*, where no agent has an incentive to deviate from their equilibrium strategy. In the most general setting (general-sum games), agents can have differing objectives, i.e. $f_i$ are not the same. In this work, we focus on cooperative games, in which case agents' interests are perfectly aligned towards a common goal i.e. $f_1 = f_2 \cdots f_N = f$. In this case, $f$ is also referred to as the game's *potential* (Monderer & Shapley, 1996).

**Diffusion Generative Models.** Diffusion generative models (DGMs) are a class of models that generate data by defining a diffusion process that transforms the data distribution into a tractable prior, and learning the time-reversal of this process. More formally, let the data distribution be $p_{\text{data}}(x)$ that is transformed into a tractable prior $p_{\text{prior}}(x)$ through a diffusion process $d\mathbf{x} = f(\mathbf{x}, t)dt + g(t)d\mathbf{w}$, where $d\mathbf{w}$ denotes the Wiener process. The time-reversal of this process is described by $d\mathbf{x}_t = \left[ f(\mathbf{x}_t, t) - g(t)^2 \nabla_{\mathbf{x}_t} \log p(\mathbf{x}_t) \right] dt + g(t)d\mathbf{w}$, where $\nabla_{\mathbf{x}_t} \log p(\mathbf{x}_t)$ is the *score* of the time-evolving distribution. This quantity is approximated by DGMs using neural networks $s_\theta$ by first sampling from the transition kernel $p(\mathbf{x}_t|\mathbf{x}_0)$ defined by the forward diffusion process, computing the score $\nabla_{\mathbf{x}_t} \log p(\mathbf{x}_t|\mathbf{x}_0)$ for $\mathbf{x}_0 \sim p_{\text{data}}$ and then regressing $s_\theta(\mathbf{x}_t)$ against $\nabla_{\mathbf{x}_t} \log p(\mathbf{x}_t|\mathbf{x}_0)$.

## 4 METHOD

In this section, we first formalize our game-theoretic view on multimeric protein docking. We then present the proposed methods and discuss their limitations (Sections 4.2-4.3). Finally, we describe the used learning architectures (Section 4.4).

**Multimeric Protein Docking as a Game.** In multimeric protein docking, proteins interact with each other through a combination of relative motions and structural changes to form the assembly structure. Notably, the space of their relative motions grows combinatorially with the number of proteins, making it challenging to compute plausible docking structures. Previous methods tackle this by modeling the relative motions between *pairs* of proteins followed by a global synchronization or a combinatorial assembly step. Instead, we argue that game theory provides a natural approach to *simultaneously* model the associated interactions and differing objectives between proteins.

In this work, we model (rigid) protein docking as a *cooperative* game, where agents' interests are aligned towards a common potential function $f$. In accordance, we view the constituted assembly structures as the underlying game equilibria, i.e. stable outcomes from which no agent has an incentive to deviate. We note that our game-theoretic framework is agnostic to whether the agents are protein chains, sub-assemblies as long as agents' incentives are modeled appropriately.

### 4.1 OVERVIEW

Modeling docking as a cooperative game requires three components – i) an appropriate definition of the action set $\mathcal{A}_i$ for each protein $i$, ii) the game potential function $f$ (either specified or learnt) capturing relevant properties of the assembly, and iii) a strategy for computing equilibria. In the rigid docking setting, we can define the action set to be the continuous space of roto-translations, i.e $\mathcal{A}_i = SO(3) \times \mathbb{T}(3)$, where $SO(3)$ corresponds to the 3D rotation group, and $\mathbb{T}(3)$ corresponds to the 3D translation group. In principle, this allows us to *simultanously* steer the proteins to equilibria configurations and thus efficiently scale to a large number of proteins.

In practice however, we do not have access to the underlying game potential $f$. To circumvent this difficulty, we describe two viable approaches in subsequent sections – i) learning an approximation of $f$ via supervised learning, using traditional physics-based scoring functions in PYROSETTA, and computing equilibria via simultaneous gradient descent (Section 4.2), and ii) viewing equilibria as samples from the Gibbs distribution of $f$, and learning a diffusion generative model over the roto-translation action spaces of all players (Section 4.3). Intuitively, the former approach corresponds to learning a differentiable analogue of traditional scoring functions to facilitate gradient-based equilibrium computation, while the latter approach can be viewed as a multi-agent extension of the denoising score-matching paradigm. A key feature of our parameterization is that the learned (reverse) diffusion can handle varying number of proteins across training and inference.

Before describing our methods, we summarize the following useful notation. As defined previously, the action space for each protein $i$ is $\mathcal{A}_i = SO(3) \times \mathbb{T}(3)$. The joint action space of all proteins $[N]$ is thus the product space $\mathcal{A} = (SO(3) \times \mathbb{T}(3))^N$. However, all our models operate directly on the protein point clouds $\mathbf{X} = [X_1, \ldots, X_N] \in \mathbb{R}^{\sum_{i=1}^N n_i \times 3}$ and residue features $\mathbf{H} = [H_1, \ldots, H_N] \in \mathbb{R}^{\sum_{i=1}^N n_i \times n_H}$. Accordingly, we can obtain the new point clouds $X_i^+$ of protein $i$ after applying rotation $R$ and translation $r$, via the transformation $X_i^+ = R(X_i - \bar{X}_i) + r$, where $\bar{X}_i$ denotes its (unweighted) center of mass.

## 4.2 Gradient-based Learning with Surrogate Potentials

**Learning the Potential Function** For each assembly structure in the training set, we generate a given number of *decoys* by sampling random rotations and translations and applying these to each protein. We then score each decoy $\mathbf{X}_j$ by its potential $f(\mathbf{X}_j, \mathbf{H})$ which in the context of this work we assume is represented by the commonly employed PYROSETTA (Chaudhury et al., 2010) energy function.[2] This constitutes our training dataset $\mathcal{D}$ for learning a surrogate game potential. To back up our choice of using PYROSETTA, we have observed assembly structures in the data to exhibit the lowest PYROSETTA energy compared to the other structures in $\mathcal{D}$ (see Figure 3 in Appendix B). To learn $f$, we employ a parametrized architecture $f_\theta$ which is invariant to permutations, translations and rotations of the input assembly, as discussed in Section 4.4.

Each protein can have a different size, number and type of atoms, and thus, the range of energy values produced for different protein assemblies can vary widely. We observed this fact hinders model generalization across different proteins when training $f_\theta$ using a simple regression (e.g., l2 error norm) objective. For this reason, we propose to train $f_\theta$ solely based on energy *comparisons*, employing the widely used ranking loss:

$$\mathcal{L}(\theta) = - \mathbb{E}_{\mathbf{X}_l, \mathbf{X}_h \sim \mathcal{D}} \big[ \log \sigma(f_\theta(\mathbf{X}_h, \mathbf{H}) - f_\theta(\mathbf{X}_l, \mathbf{H})) \big] \tag{1}$$

where $\mathbf{X}_h, \mathbf{X}_l$ are random pairs of decoys such that $f(\mathbf{X}_h, \mathbf{H}) > f(\mathbf{X}_l, \mathbf{H})$, i.e. $\mathbf{X}_l$ has lower (true) energy than $\mathbf{X}_h$, and $\sigma$ is the logistic function. This is a widely-used loss in preference-based learning and corresponds to maximum likelihood under a Bradley-Terry reward model, see e.g. (Bradley & Terry, 1952; Rafailov et al., 2023).

**Equilibrium Computation via Gradient-Based Learning** Once $f_\theta$ has been trained, we can compute assembly structures by randomly initializing each protein and updating their roto-translation actions via simultaneous gradient descent. In a potential game, this ensures convergence to equilibria (which are local minima of $f_\theta$), but in case of general-sum games more sophisticated update rules could be employed, see e.g., (Balduzzi et al., 2018). A caveat of PYROSETTA scores, though, – which is also inherited by $f_\theta$ – is that assembly structures where constituent proteins are far away (in 3D space) from each other also have low energy, which is undesirable as a docking outcome. This was also observed empirically in our early experiments. To discourage this behavior, we thus add a distance-based penalty $d(X_i, X_J) := \text{ReLU}(d_{\text{res}}(X_i, X_J) - d_{\text{ths}})$ which computes the minimum Euclidean distance between any pair of residues belonging to proteins $i$ and $j$ and penalizes the agents only when this distance is greater than a threshold $d_{\text{ths}}$ (in our experiments, this was 5Å). Overall, the action updates for proteins' reads as:

$$(R_i^{t+1}, r_i^{t+1}) = (R_i^t, r_i^t) + \eta^t \cdot \nabla_{(R_i, r_i)} \Big[ f_\theta(\mathbf{X}^t, \mathbf{H}) + \lambda \cdot \sum_{j \neq i} d(X_i^t, X_j^t) \Big], \quad i \in [N], \tag{2}$$

where $\eta^t$ is a (decreasing) learning rate, $\nabla_{(R_i, r_i)}$ is the Riemannian gradient (i.e. living in the tangent space of $SO(3)$ and $\mathbb{T}(3)$) and $\lambda$ is a tunable weight. In practice, the above updates need not necessarily be simultaneous as one could also update proteins sequentially in a round-robin fashion.

**Discussion** On one hand, the above approach is quite general and flexible: The discussed gradient-based update scheme can be carried with any potential function (also fully specified, and not necessarily trained) as long as it is differentiable. Moreover, it can also be easily extended to the case where each protein has its own objective $f_i$ by adding penalty terms that are specific to each protein. In addition, while not the focus of this work, $f_\theta$ can also be used beyond equilibrium computation, e.g. to characterize the associated energy landscape (see Jin et al. (2023)). On the other hand, though, it heavily relies on the supervision signal used to train $f_\theta$, which may not accurately reflect the true underlying game structure. In the next section, we propose an alternative approach that circumvents the need to utilize this supervision and learns solely from the observed assemblies.

## 4.3 Diffusion Generative Models over the Proteins' Action Spaces

Section 4.2 views assembly structures as local minima of $f$ wrt roto-translation gradients of all proteins. An equivalent characterization of the assembly structures is interpreting them as sample(s)

---

[2]The approach is evidently more general and any scoring/ranking function can be applied. We stick to PYROSETTA because it is the one we use in our experiments.

from the mode(s) of the Gibbs distribution $p \propto \exp(-f)$. Furthermore, it is easy to see that for any protein $i$, $\nabla_{(R_i, r_i)} \log p = -\nabla_{(R_i, r_i)} f$ where $\nabla_{(R_i, r_i)} \log p \in T_{R_i} SO(3) \oplus T_{r_i} \mathbb{T}(3)$ are the Riemannian gradients in the tangent space of $SO(3)$ and $\mathbb{T}(3)$ at $R_i, r_i$ respectively. This implies an equivalence between the roto-translation scores of $p$ and the roto-translation gradients wrt $f$, and connects equilibrium computation to sampling for cooperative games. We can thus frame rigid multimeric docking as a generative modeling problem over the joint roto-translation action space $\mathcal{A}$ using the framework of diffusion generative models. Intuitively, this corresponds to learning $f$ implicitly through the (perturbed) scores of $p$.

**DGMs over the joint action space $\mathcal{A}$**   De Bortoli et al. (2022) showed that the theoretical framework of DGMs for standard Euclidean spaces (Section 3) also holds for compact Riemannian manifolds with minor modifications, with the drift function $f(\mathbf{x}_t, t)$ and the score $\nabla_{\mathbf{x}_t} \log p(\mathbf{x}_t)$ being elements of the tangent space, and the reverse diffusion corresponding to a geodesic random walk on the manifold. Furthermore, following the arguments in Corso et al. (2023); Rodolà et al. (2019); Yim et al. (2023) – since $\mathcal{A}$ is a product space (more precisely manifold), the forward diffusion process proceeds independently on each component manifold, and the tangent space is a direct sum - $T_{(\mathbf{R},\mathbf{r})}\mathcal{A} = \bigoplus_{i=1}^{N} T_{R_i} SO(3) \oplus T_{r_i} \mathbb{T}(3)$, where $\mathbf{R} = (R_1, R_2, \cdots, R_N)$ and $\mathbf{r} = (r_1, r_2, \cdots, r_N)$. This allows us to apply standard denoising score matching objectives (Song & Ermon, 2019; Song et al., 2020) independently for each component. Moreover, to ensure translation invariance, we keep one protein fixed with its COM at the origin, and define the diffusion model only over the joint action space of the remaining $N - 1$ proteins.

**Diffusion Processes over $SO(3)$ and $\mathbb{T}(3)$**   For both $SO(3)$ and $\mathbb{T}(3)$, we use the Variance Exploding SDE (VE-SDE) formulation (Song et al., 2020) to define the forward noising process $d\mathbf{x}_t = \sqrt{\frac{d[\sigma^2(t)]}{dt}} d\mathbf{w}$, where $\sigma$ is defined as an exponential schedule $\sigma(t) = \sigma_{\min}^{1-t} \sigma_{\max}^{t}$, with appropriate values for $SO(3)$ and $\mathbb{T}(3)$. The transition kernel on $SO(3)$ is the $IGSO(3)$ distribution (Nikolayev et al., 1997; Leach et al., 2022), which can be sampled in the axis-angle parameterization of $SO(3)$ as described in (Corso et al., 2023; Yim et al., 2023), by sampling a unit vector $\hat{\omega}$ uniformly and a $\omega$ according to:

$$p(\omega) = \frac{1 - \cos\omega}{\pi} f(w) \quad \text{where} \quad f(\omega) = \sum_{l=0}^{\infty} (2l+1) \exp(-l(l+1)\sigma^2) \frac{\sin((l+1/2)\omega)}{\sin(\omega/2)} \quad (3)$$

The score of the transition kernel is given by $\left(\frac{d}{d\omega} \log f(\omega)\right) \hat{\omega}$, which can be precomputed by truncating an infinite sum. Sampling from the transition kernel can be accomplished by interpolating the CDF of $p(\omega)$. For $\mathbb{T}(3)$, the transition kernel is simply the standard Gaussian with variance $\sigma^2(t)$ and the score of the transition kernel is simply $-r_t^2/\sigma^2(t)$.

**Computing Equilibria**   Under the DGM paradigm, computing equilibria is equivalent to sampling using reverse diffusion guided by the learnt score. The reverse diffusion, as mentioned above, corresponds to a discretized geodesic random walk on the joint action space $\mathcal{A}$.

We note that connections between sampling, SDEs and games (Chen et al., 2021; Liu et al., 2022) have also been explored in previous works, assuming either an infinite-player setting or a homogenous player one. In contrast, our approach is designed for a finite-player setting and places no restriction on player types. Furthermore, as discussed in Section 4.4, the parameterization of our score-network allows us to handle a varying number of players across training and inference.

## 4.4 ARCHITECTURES

In this section, we summarize the data representations and network architectures utilized by the proposed approaches. We defer more formal descriptions and additional details to Appendix A.

All our architectures take as inputs the protein point clouds $\mathbf{X}$ and residue features $\mathbf{A}$. The point clouds are then used to dynamically build graphs based on appropriate distance cutoffs between residues of the same and different proteins. Residue representations are then learnt using message-passing. The potential network and the score network share similar architectures for the message-passing layers, with the only differences in the output layers. The potential network predicts a

$SE(3)$-invariant scalar, while the score-network predicts an output in the tangent space $T_{(\mathbf{R}, \mathbf{r})}\mathcal{A} = \bigoplus_{i=1}^{N} T_{R_i} SO(3) \oplus T_{r_i} \mathbb{T}(3)$, where all outputs are $SE(3)$-equivariant vectors.

**Residue Representations**   Residue representations are learnt using message-passing layers based on tensor products. These message-passing layers are based on $SE(3)$ convolutions using tensor products, as implemented in the `e3nn` library (Geiger & Smidt, 2022). We use separate message-passing layers for edges within the same protein and between different proteins. These messages are then aggregated to produce scalar and vector representations for each residue.

**Potential Network**   The output of the potential network $f_\theta$ is a $SE(3)$-invariant scalar, which can be interpreted as the energy of the system. We first generate edge representations by concatenating the scalar components of the corresponding residue representations, which are then passed to fully connected layers followed by a mean-pooling step to compute edge energy contributions. The edge and residue contributions (computed similarly) are added up to predict the energy.

**Score Network**   For each protein $i$ (except the fixed protein), the score network $s_\theta$ takes the corresponding learnt residue representations as input and computes the rotational and translational scores (i.e. two $SE(3)$-equivariant outputs) via a tensor-product convolution with the COM of protein $i$, as done in Corso et al. (2023). This parameterization allows the score network to handle varying number of agents across training and inference, and could be of independent interest.

## 5 EXPERIMENTS

**Datasets**   We use two datasets in our experiments: Docking Benchmark 5.5 (DB5.5) and the Database of Interacting Protein Structures (DIPS). DB5.5 is a standard dataset used in benchmarking docking methods and contains 253 assembly structures. DIPS is a larger dataset containing 42826 assembly structures, but only consists of single-chain proteins. While DB5.5 has been traditionally used in the context of binary protein docking, many examples consist of proteins made up of multiple chains, allowing us to evaluate DOCKGAME for multimeric docking. We use the same splits for DIPS and DB5.5 datasets as Ganea et al. (2022) for our experiments.

**Experimental Setup**   Following Ganea et al. (2022), we first train our models on the DIPS dataset, and finetune it on the DB5.5 dataset. Unlike Ganea et al. (2022), however, we train our models at the granularity of protein chains. For the DIPS dataset, this implies no difference as all examples comprise two single-chain proteins. However, on the DB5.5 dataset, the examples comprise proteins with 2-8 chains. To the best of our knowledge, this is the first usage of the DB5.5 dataset for multimeric docking experiments. More experimental details can be found in Appendix B, with code available at `https://anonymous.4open.science/r/iclr24-dockgame`.

**Baselines**   We compare DOCKGAME against traditional binary docking methods ATTRACT, CLUSPRO, PATCHDOCK, the traditional multimeric docking method MULTI-LZERD, and recent deep learning methods for binary docking, EQUIDOCK and DIFFDOCK-PP. All baselines except MULTI-LZERD are binary docking methods. Among other multimeric docking methods, we do not include any comparisons to ALPHAFOLD-MULTIMER and SYNDOCK as the DB5.5 dataset is part of ALPHAFOLD-MULTIMER's training set, while SYNDOCK has no open-source implementation available at the time of writing. More details regarding the baselines can be found in Appendix B.4.

**Evaluation**   DB5.5 has traditionally been used for benchmarking binary docking methods, where the task is to predict the best relative orientation (or a set thereof) between two proteins (referred to as ligand and receptor). For such binary docking baselines, we utilize the evaluations as provided with code in Ganea et al. (2022). However, many examples in DB5.5 consist of proteins that are subassemblies of *multiple* chains. This fact is neglected by the aforementioned binary docking, where the relative configuration between all chains in the subassembly is assumed constant and already specified. Here, we test DOCKGAME on the significantly harder multimeric docking setting, where the relative configuration even between chains in the same subassembly needs to be inferred. Notably, this task has many more degrees of freedom than its binary counterpart, and, to the best of our knowledge, it has not been considered in the literature.

Table 1: **Assembly Structure Prediction on DB5.5 Dataset**. For the first five baselines, the inputs consist of two proteins, with the goal of predicting their assembly structure (binary docking). Instead, DOCKGAME and MULTILZERD take as input constituent protein chains, and are faced with the harder *multimeric* docking task. DOCKGAME-E refers to the DOCKGAME model with the learnt potential function (Section 4.2), while DOCKGAME-SM, refers to the DOCKGAME model with the learnt score network (Section 4.3). We use both methods to compute multiple (20 and 40, resp.) assemblies for each complex. The rule "(X, filtered by TM-score)" implies that, among the X generated assemblies, we identify the one with the highest TM-score to the test data and consider all other generated assemblies within a 0.05 TM-score absolute difference. This copes with the fact that, while DOCKGAME (and DIFFDOCK-PP) can generate multiple plausible equilibria, DB5.5 test set contains only a single assembly structure per example. The rule "(X, best C-RMSD)" implies that, among the X generated assemblies, we consider the one with the smallest C-RMSD.

| Method | Avg. Runtime | C-RMSD | | | TM-score | | |
|---|---|---|---|---|---|---|---|
| | (in [s]) | Mean | Median | Std | Mean | Median | Std |
| **Binary Docking** | | | | | | | |
| ATTRACT | 570 | 10.63 | 12.71 | 10.05 | 0.8317 | 0.8256 | 0.1668 |
| CLUSPRO | 15507 | 8.26 | 3.38 | 7.92 | 0.8318 | 0.8938 | 0.1535 |
| PATCHDOCK | 3290 | 18.01 | 18.26 | 10.12 | 0.7270 | 0.7335 | 0.1237 |
| EQUIDOCK | 5 | 14.72 | 14.1 | 5.3 | 0.7191 | 0.7107 | 0.1078 |
| DIFFDOCK-PP | | | | | | | |
| (40, filtered by TM-score)* | 80 | 17.19 | 16.29 | 6.79 | 0.7086 | 0.7312 | 0.1142 |
| (40, best RMSD) | 80 | 12.81 | 11.79 | 4.61 | 0.7014 | 0.6773 | 0.1125 |
| **Multimeric Docking** | | | | | | | |
| DOCKGAME-E | | | | | | | |
| (20, filtered by TM-score) | 182 | 19.28 | 17.74 | 7.37 | 0.6714 | 0.6820 | 0.1543 |
| (20, best C-RMSD) | 182 | 14.18 | 11.54 | 9.24 | 0.6182 | 0.6257 | 0.1795 |
| DOCKGAME-SM | | | | | | | |
| (40, filtered by TM-score) | 157 | 14.12 | 9.44 | 4.75 | 0.7173 | 0.7773 | 0.1419 |
| (40, best C-RMSD) | 157 | 8.73 | 8.72 | 4.68 | 0.7246 | 0.7681 | 0.1578 |
| MULTI-LZERD | 82753 | 21.23 | 20.89 | 6.82 | 0.6312 | 0.6266 | 0.1352 |

**Metrics** Docking methods are commonly evaluated by the Complex Root Mean Squared Deviation (C-RMSD) metric, which measures the deviation between the predicted and ground truth assembly structure after Kabsch alignment (details in Ganea et al. (2022)). We also utilize TM-score (Zhang & Skolnick, 2005) as an alternative metric to quantify structural alignment.

A fundamental evaluation problem for DOCKGAME and DIFFDOCK-PP is that they can both generate multiple plausible equilibria, while the test set of DB5.5 only has a single assembly structure per example. We thus compute summary statistics (as described by Mean, Median and Std. deviation of C-RMSD and TM-score) on a filtered set of the predicted assemblies, constructed in two ways:

- Identifying the predicted assembly with the highest TM-score (as an effective measure of structural alignment) to the ground truth, and considering all predicted assemblies within a TM-score (to ground truth) radius of 0.05. We adopted this heuristic to extract predicted assemblies close to the equilibrium present in the data, and filter out different (but still potentially plausible) equilibria generated by DOCKGAME.

- Identifying the predicted assembly with the lowest C-RMSD to the ground truth.

**Results and Discussion** Despite the harder multimeric docking task, DOCKGAME-SM, trained only with assembly structures, achieves comparable performance to binary docking methods on both the C-RMSD and TM-score metrics (Table 1). In particular, the median C-RMSD (more robust to outliers) for DOCKGAME-SM across both filtered sets is better than all baselines except CLUSPRO (which however requires a significantly higher runtime). Furthermore, when compared with the traditional multimeric docking method MULTI-LZERD, both DOCKGAME-E and DOCKGAME-

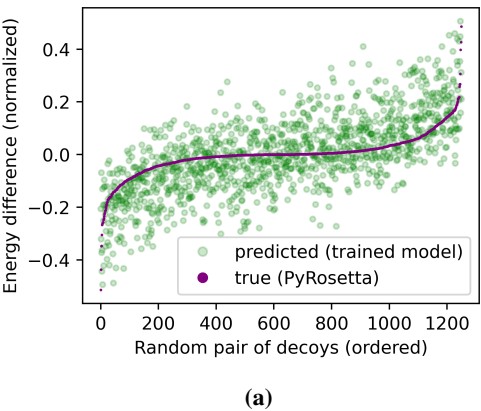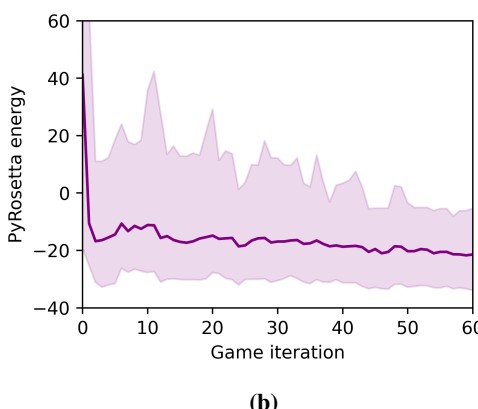

|       |       |
|:-----:|:-----:|
| **(a)** | **(b)** |

Figure 2: **(a)** Learned vs. True PYROSETTA energy differences, over 50 random decoy pairs for each complex in DB5.5 test. **(b)** PYROSETTA energetics during game rounds. Median, 25-75th percentiles, over 20 games for each complex in DB5.5 test.

SM achieve significantly better performance (both in terms of C-RMSD and TM-score), with almost 3 orders of magnitude faster runtimes.

To better assess the performance of DOCKGAME-E, in Figure 2(a) we plot the energy differences predicted by the trained potential network, versus the respective differences computed by PYROSETTA. We can see that a good approximation of the PYROSETTA energy is achieved from our training set containing only 10 pairs of roto-translation perturbed structures per example. Furthermore, in Figure 2(b) we observe that DOCKGAME-E reduces the PYROSETTA energetics during gradient-based equilbrium computation as desired. This demonstrates the utility of DOCKGAME-E as a faster, differentiable alternative for traditional scoring methods. However, the improved performance of DOCKGAME-SM relative to DOCKGAME-E highlights that PYROSETTA might not offer the best supervision signal if the goal is predict assembly structures close to the ground truth. DOCKGAME-SM might be the more preferred alternative, especially in data-rich settings.

Finally, we highlight that while DOCKGAME can naturally predict different assembly structures (equilibria) for every example, we utilized the aforementioned filtering schemes to compare these to the only assembly structure present in the dataset. An exciting avenue for future work would be to explore ways to assess the plausibility of multiple generated equilibria, which is currently lacking.

# 6 CONCLUSION

In this work, we presented DOCKGAME, a novel game-theoretic framework for rigid multimeric protein docking. We view protein docking as a cooperative game between protein chains, described by a underlying potential whose equilibria constitute the assembly structures. To learn the unknown game potential and compute equilibria, we propose two approaches – i) a supervised approach using traditional scoring functions to learn a surrogate potential function, ii) a self-supervised approach based on diffusion generative models, that learns solely from observed assembly structures. Both our methods can efficiently decouple the combinatorial docking space into individual protein's action spaces. Moreover, our game-theoretic framework allows us to compute multiple assembly structures as plausible equilibria. We evaluated DOCKGAME on the harder multimeric docking task (compared to binary docking baselines) using the DB5.5 dataset, and achieve comparable (or better) performance on C-RMSD and TM-score metrics, while exhibiting significantly faster runtimes.

While the focus of this work is on cooperative games, the presented ideas are general and can be extended to general-sum games, allowing us to model the flexibility of proteins, which is missing in this work. Potential ways to do so would be addition of protein structure specific penalty terms, or combining diffusion bridges (Holdijk et al., 2022; Somnath et al., 2023) with the proposed diffusion generative model. Other interesting avenues of future work would be in the evaluation of multiple equilibria, and developing new metrics in this regard.

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
