# OpenReview forum: "DockGame: Cooperative Games for Multimeric Rigid Protein Docking"
_ICLR.cc/2024/Conference — Submitted to ICLR 2024_

### Official Review · Reviewer_tNw5 · 2023-10-26

**Soundness:** 1 poor
**Presentation:** 3 good
**Contribution:** 2 fair
**Rating:** 3
**Confidence:** 5

**Summary:**

In the presented work, the authors propose two methods for solving the multimeric rigid protein docking problem. Especially the extension of prior work for *multimeric* docking is interesting. The work is motivated by game-theoretic principles for stable equilibria in potential games. The first method finds the stable equilibria by learning a surrogate model of the molecular potential that can be minimised using gradient descent. Additionally, to achieve sample diversity, the method uses multiple initialisations for the procedure. A second proposed method extends on diffusion models for rigid protein docking to allow for multimeric complexes.

**Strengths:**

**Originality and significance**: The paper extends on the standard rigid protein docking problem to also cover multimeric complexes. This is an important step forward and therefore the work has originality and significance.

**Quality**: The proposed method is compared to a large number of other methods including both recent ML based approaches and more search intensive traditional methods.

**Clarity**: The paper is very well written with clear exposition of the problem that it aims to solve as well as some of the main background information needed to follow along.

**Weaknesses:**

Unfortunately, the paper has a number of weaknesses that I believe prevent the paper from being accepted at this time. I've included detailed comments about these weaknesses (and also positive points) in the comments section below.

**In short, the main weaknesses are the following:**
- The game-theoretic approach is not well enough formalised. This makes it hard to judge to what extend the game-theoretic view on the multimeric rigid protein docking problem helps in the understanding of the problem.
- Some of the modelling choices, especially the use of a surrogate model for the molecular potential, is not sufficiently motivate. At this time, I am unsure why the surrogate model is needed.
- The evaluation presented ignores one of the major claims of the paper; allowing the sampling of an ensemble of samples. As a result of this, it is hard to judge the main contributions of the paper.

**Questions:**

Please find my detailed list of comments, ordered by section, below.

Where needed, I indicated that something might be a breaking issue. These issue will have to be either clarified or rectified before I will consider increasing my vote to accept.

**Abstract**
1. I find the abstract to be very well written and it definitely peaked my interest.
2. One question that popped up in my mind several times while reading the abstract (and the similarities became more clear further in the paper) is the difference between the *game potential* and the *molecular potential* that defines, to some extend, the possible configurations in a physical sense. I believe that this is what the paper refers to with "physics-based energy function", but this was slightly confusing.


**Introduction:**
1. The first two paragraphs of the introduction are very well written and do a great job introducing the problem setting and the innovation of the paper.
2. As indicated above, I believe the paper does not properly set out the relation between the potential function, in game theoretic sense, and the molecular potential. This issue pops up in the third paragraph of the introduction where the paper mentions that the "final assembly structure(s) constitute stable equilibria". The paper should relate the stable equilibria, in the game theoretic sense, with meta-stable states of the protein-protein complexes.
3. When the author mention "a differentiable analogue of traditional (and black-box) physics-based scoring function" they should make clear that this refers to the an empirical force field.
4. In the introduction the paper should cite the original Diffusion model papers in the last paragraph before listing the contributions.

**Related Work**
1. Ketara et al., which introduced the DiffDock-pp method, generates an ensemble of possible complexes. This makes the method very similar to the proposed method in the sense that they both are able to compute multiple equilibria. I understand that Ketara et al. do introduce a method to rank these, but by definition it is able to define a distribution of samples. This similarity should be made more clear in the related work discussion of this work.

**Preliminaries and Background**
1. The problem definition is very well written and adds a lot to the readability and clarity of the paper.
2. I would like to ask the authors to clarify the cost function / potential introduced in the paragraph title "Game Theoretic Concepts". From my understanding, a potential function (or as the authors first call it, a cost function) is an abstraction of the payoff function, and as such, should really be defined as a function over the current states of the agent. It is possible to define the potential function using sequences of actions, or strategies, but this is only possible with a fixed starting state. Importantly, when introducing the used potential function in section 4.2, it is, in contrast to the original background discussion, actually defined over the state space (in the form of protein point clouds). This confusion is a breaking issue at the moment and should be clarified.
3. In general, the section introducing the Game-Theoretic Concepts highlights to me that the Game-theoretic aspects of the proposed method is not very strong. As it stands, the paper does not relate the potential function, in the game theoretic sense, to the molecular potential and does thus not show that the stable equilibria, in the game-theoretic sense, are related to local minima of the molecular potential. In general, as highlighted more clearly, the game theory aspect could be taken out completely by clearly stating that the potential f is a learned approximation of the true molecular potential.
4. For the definition of the time-reversal, the definition here should be kept consistent with the forward process by defining it as $$ d\mathbf{x} = [f(\mathbf{x, t}) - g(t)^2\nabla_\mathbf{x} \log p_t(\mathbf{x})]dt +g(t)d\bar{w}$$ where $\bar{w}$ is the time-reversed Wiener process. This is also the formulation used in the Diffusion model papers cited elsewhere in the paper. It is important that the distribution $p$ is uniquely defined for every $t$.
5. Talking about the cited diffusion model papers, the introduction in the" Diffusion Generative Models" paragraph is very minimal (eg. it does not introduce the score matching objective used). To make the paper more accessible, the authors should add some of the papers cited in later sections of the paper here.

**Method**
1. When discussing multimeric protein docking, the authors should be careful that they clarify that we are talking about the *rigid* version of this problem. In the current version of the paper this is not always clarified.
2. When stating "Previous models tackle this [..]" the paper should include citations to the specific previous methods that do this.
3. As mentioned before, when the paper mentions just above starting section 4.1 "game equilibria, i.e. stable outcomes from which no agent has an incentive to deviate", the paper should justify that this is a correct objective to aim for by showing, formally, that these game equilibria are located at local minima of the molecular potential landscape.
5. In the second paragraph of section 4.1 the authors state "we do not have access to the underlying game potential f" and while this is true to some extend (using DFT at this scale is not possible) we do have very good approximations available in the form of empirical force fields. This misunderstanding becomes more clear in section 4.2 where the authors introduce their gradient-based learning with surrogate potentials. It is unclear to me why the authors require the training of a surrogate model and don't use the energy minimisation functionality from Rosetta to find the equilibria for the randomly initialised proteins. This is a breaking issue for me and should be clarified by the authors in their response.
6. The last paragraph of section 4.1 helps a lot in the formalisation of the proposed methods. I applaud the authors for taking the space to carefully introduce their notation.
7. A second breaking issue in this section is the formulation of the loss function used to train the surrogate model. My observation here is that while the use of the ranking loss retains relative ordering of the different sampled complexes in terms of energy, it does not accurately reflect the gradient of the energy surface. This limits the applicability of the method for deriving of thermodynamical properties in possible downstream applications due to possible oversampling of high energy complexes. Since deriving thermodynamical properties is not an objective of the paper, this is a not necessary an breaking issue, however, it does conflict with a later statement in the discussion paragraph of section 4.2 where the paper states "$f_\theta$ can also be used beyond equilibrium computation e.g. to characterise the associated energy landscape (...)".
8. The paper states "in case of general-sum games more sophisticated update rules could be employed [...]", i'm unsure how this translates to the problem at hand. I would appreciate it if the authors could include an example of how the protein docking problem could be defined as a general-sum game. Similarly, i'm unsure what the physical realisation of adding penalty terms for each protein would be as the authors discuss in the discussion of this section.
9. In section 4.3 the paper states "An equivalent characterization of the assembly structures is interpreting them as sample(s) from the mode(s) of the Gibbs distribution". This highlights my concern with the game-theoretic framework. The target docked complexes are *exactly* the modes of the Gibbs distribution if we defined f to be the molecular potential.
10. Section 4.3 is really well written.
11. In section 4.4 the paper switch from using $\mathbf{H}$ to $\mathbf{A}$ for referring to the residues features. Personally, I find this switch confusing and would like to ask the authors to clarify why this change was made. While using H was well defined, using A for the features conflicts with the notation of the agents actions.
12. The architecture definition clarifies that graphs are used to model the interactions between both the residues of the same and different proteins. Considering that the paper focusses on *rigid* docking, I'm unsure why we need internal interactions. It would be great if the authors could clarify their reasoning regarding this.

**Experiments**
1. Am I correct in assuming that DIPS was only used for training the model and not for evaluation? In the "Datasets" paragraph the paper mentions "We use the same splits for the DIPS and DB5.5 datasets as Gana et al. for our experiments", but I don't see any evaluation of the DIPS dataset.
2. For the comparison with DiffDock-PP. Given that a similar manifold diffusion model is used, how considerable is the difference between the proposed model and DiffDock-PP? And related, does that difference explain the difference we see between DockGame -SM and DiffDock-PP?
4. For table 1, due to the large amount of model evaluated, it is hard to compare the differences. It would be beneficial if the authors could highlight the best scores for each metric.
5. Also related to table 1, based on the presented standard deviations, it seems like all methods very much overlap in their scores. Alternatively, I could see understand that the standard deviation refers to the samples in the filtered set of the predicted assemblies, but in that case I do not understand how that works for the C-RMSD score.  It would be good if the authors could comment on this.
6. Related, for the TM-Score cutoff of 0.05, it would be good if the paper could comment on how this relates to the identification of the different meta-stable states. If this threshold is set such that all accepted samples are from the same meta-stable state, I would expect all accepted samples to be exactly the same as the only source of stochasticity is the complex initialisation.
7. Related to the previous comment about the diversity of the accepted samples, the authors state "an exciting avenue for future work would be to explores ways to assess the plausibility of multiple generated equilibria, which is currently lacking". I strongly believe that looking at the sample diversity should be already addressed in the present paper. This is a breaking issue for me as there is currently no way to assess their contribution regarding the use of sampling multiple equilibria. For this purpose I would expect at-least a qualitative visual exploration, but a quantitative exploration of the diversity would be preferred.

---

> ### Author Response · Authors · 2023-11-22
> **Author Response - 1/4**
>
> **Abstract, Q2** :Thank you for the comment.
>
> The term molecular potential is typically used to refer to the function that characterizes the energy landscape of different molecular configurations.
>
> The game potential, on the other hand, is not just restricted to energetics. While we do agree that in the context of the supervised learning section of the paper, they are the same, in general this need not be the case. The game potential could just as well represent a function that characterizes some other property such as toxicity or solubility for instance, without any modifications to the strategy to compute equilibria.
>
> Our intention while writing was to not show how the molecular potential resembles the game potential, but instead to map out different components used in defining a game to appropriate aspects of the rigid multimeric docking problem.
>
>
> **Introduction, Q1**:
> Thank you for the suggestion. We are happy to emphasize this point more in the updated version of the paper, and more formally connect the notion of equilibria to assembly structures.
>
> **Introduction, Q2**:
> Thanks, we will incorporate this point.
>
> **Introduction, Q3**: Below we include the relevant statement from the Related Work section:
>
> “The closest related work to ours are (Corso et al., 2023; Ketata et al., 2023), of which our diffusion generative model for the cooperative game can be interpreted as a multi-agent extension.”
>
> We believe this statement sufficiently captures the intent of the reviewers comment, especially since we do not emphasize the confidence model used in DiffDock-PP to rank samples and actually refer to them as a generative model.
>
> Second, the diffusion generative model, while an obvious extension of DiffDock-PP, provides a general recipe for defining diffusion models over (continuous) action spaces in cooperative games, connecting diffusion models with the multi-agent community, and allowing either community to benefit from advances in the other.
>
> Third, the connections between sampling in diffusion models and equilibrium computation in (cooperative) games are interesting in their own right. In the multi-agent literature, connections between sampling and equilibrium computation have been explored largely in the context of mean-field games (where one has infinite agents), or games with a large but finite but large number of homogenous agents.
>
> In contrast, our diffusion model does not place an assumption of the types of players, and provides a general recipe to define diffusion models over the (continuous) action spaces of all players in the cooperative game, when one only has access to equilibria and not the underlying game structure.

---

> ### Author Response · Authors · 2023-11-22
> **Author Response - 2/4**
>
> **Preliminaries and Background, Q2**:  In the multi-agent and game-theory communities, the payoff function for each player in the game depends on the actions (or strategies) adopted by all players, and we stick to the same definition. However, there are some subtleties that apply when working with point clouds in Euclidean spaces with rotations and translations as actions for each player.
>
> First, both rotations and translations are **groups**, and applying a rotation and translation to an element in the Euclidean space corresponds to the formal notion of **group action**. The notion of **group action** is what allows us to forget about the fixed starting state, as can be seen from the following:
>
> Let $x$ be a fixed starting state, and let $R_a, R_b$ be rotations, whose action on $x$ results in $x_a, x_b$ respectively. As the set of rotations form a group, there exists a rotation $R_bR_a^-1$, which when applied to $x_a$ corresponds results in $x_b$, since it first reverses $x_a$ back to $x$, and then applies $R_b$ to $x$. Thus the group action of $R_b$ on $x$ produces the same result as $R_bR_a^-1$ on $x_a$, allowing us to “forget” the location of the starting state $x$.
>
> Second, providing the full 3D structure of the proteins, rather than abstract elements from the action space for each player, allows the model to reason about physical interactions and better generalize to unseen assemblies. This viewpoint is not new, and was also adopted in DiffDock & DiffDock-PP.
>
> **Preliminaries and Background, Q3**: First, the game-theoretic aspect of the paper is not just concerned with the presence of an underlying potential, but also on the strategy used for computing equilibria when given such a potential and appropriate action spaces. Since the action spaces are continuous, gradient-based methods for computing equilibria can be applied. The game-theoretic perspective, thus allows to formulate multimeric docking as a gradient-based optimization problem instead of the traditional combinatorial assembly paradigm.
>
> Second, in the context of multimeric docking, we do not have access to the underlying potential (or payoffs in case of general-sum games) but only equilibria samples. Thus, to compute equilibria, we need to either specify or learn an appropriate potential. In the multi-agent literature, this is referred to as the inverse multi-agent RL problem (the RL is not applicable in our setting). Both approaches are designed to specifically tackle this problem of inferring the potential before computing equilibria.
>
> We do agree that in the context of cooperative games, equilibria correspond to local minima and equilibrium computation is equivalent to minimizing the potential, this does not make the adopted strategies any less game-theoretic. We have also given an example in a later answer for the case of general-sum games, where gradient-based algorithms for equilibrium computation are just as applicable.
>
> We are however happy to include a justification for why the assembly structures constitute equilibria wrt an appropriate potential, and make a more formal argument in this regard.
>
> **Method, Q1**: Thanks, we have tried our best to ensure this is mentioned in all the relevant places but are happy to include it in other parts of the paper accordingly.
>
> **Method, Q2**: Thank you, we will include relevant citations while referring to previous methods in the text.
>
> **Method, Q3**: Thank you, we will include a more formal characterization of equilibria and a justification of why the objective is an appropriate one to aim for in the updated version(s) of the paper.
>
> **Method, Q4**: There are multiple reasons for this:
>
> - Rosetta has limited support for any computations on GPU and their docking protocols are only designed for binary docking. Any extension to multimeric docking would require writing a custom protocol that follows the traditional paradigms of pairwise docking + combinatorial assembly, which is precisely the disadvantage we highlighted in the Introduction. This would also make multimeric docking very slow, as evidenced by the time taken by MultiLZerd for the same multimeric docking problem (Table 1). In contrast, DockGame can run on GPU and has much faster runtimes.
>
> - Diffusion models can additionally be augmented with a guidance term that can be used to steer the diffusion towards targets of interest. The potential network thus learnt in Section 4.2 can be used as additional guidance term. This is easy from an implementation perspective when all the computations can be done on GPU, which is not possible with Rosetta or other docking protocols.
>
> - While we use the PyRosetta energetics in this paper to learn an approximation, we could just as easily use a pretrained property predictor as the game potential f. The goal was to showcase a proof-of-concept for learning the game potential, and using it to compute equilibria rather than the *choice* of the functions used to generate supervised data.

---

> ### Author Response · Authors · 2023-11-22
> **Author Response - 3/4**
>
> **Method, Q6**: We are happy to take the statement regarding $f_{\theta}$ out of the paper. However, we respectfully disagree with the statement on the accuracy of the gradient landscape. The equilibria are computed by taking gradients of the learnt potential function, and as shown in Fig 2(b), the model is able to find assembly structures with low energy according to PyRosetta as the game progresses, so the model is indeed able to learn (in expectation) accurate gradients through pairwise ranking comparisons.
>
> **Method, Q7**: A general-sum game is defined by a different payoff function for each player. In the case of proteins, the payoff could be for instance the common payoff (as is learnt currently), dictating some properties of the assembly (like energetics or other task-specific properties) plus an additional weighted (weighted per protein) penalty term that minimizes the deviation from the native structure. We do note that general-sum games are more applicable in the context where we have both the unbound and bound structures of the proteins, rather than the setting considered here where we only focus on rigid bound structures.
>
> The intuition behind the payoff functions is that the unbound (or native) structures of the protein are stable when said protein is considered in isolation, but this structure might not be optimal when forming the assembly. There is thus a tradeoff between the selfish objective of the protein (maintaining its own structure closer to the native one), and the common cooperative objective of forming an assembly.
>
> A combination of a common cooperative objective with a weighted penalty term provides different payoff functions for each protein, thus constituting the general-sum game formalism.
>
> **Method, Q8**: Thank you for the comment. We hope the following rephrasing will help clarify any concerns:
>
> If f is defined as the molecular potential, the corresponding Gibbs distribution has a higher density at assembly structures with lower potentials. When drawing samples from this distribution, we are more likely to observe assembly structures that have higher density. These are the samples over which we learn the diffusion generative model.
>
> **Method, Q10**: Thank you for pointing this out -- this should indeed be $\textbf{H}$ in section 4.4 and not $\textbf{A}$.
>
> **Method, Q11**: Internal interactions allow residues to reason about their local environments and represent higher-level properties (say a hydrophobic patch on one protein for instance). In combination with external relations, this allows the model to reason about properties important for assembly formation (for e.g. bringing the hydrophobic patches of two proteins closer together so they can be buried in the binary assembly) and predict the rotational and translational scores accordingly.
>
> Without the use of internal interactions, the model would have a much harder time connecting protein specific properties to assembly properties and model training will be slower.

---

> ### Author Response · Authors · 2023-11-22
> **Author Response - 4/4**
>
> **Experiments, Q1**: The DIPS dataset was only used for training, followed by finetuning on the DB5.5 dataset, following the procedure as in Ganea et al. While their goal was to evaluate *binary* docking, we wanted to evaluate *multimeric* docking. The DIPS dataset only consists of 2 protein (chain) examples, while the DB5.5 dataset had examples with multiple chains in the same protein, which allowed us to evaluate our method at the level of multimeric docking.
>
> **Experiments, Q2**: For the case of rigid bodies, DockGame can be interpreted as a multi-agent extension of DiffDock-PP.
>
> From a training procedure, there is no difference on the DIPS dataset since it only has examples with two chains. However, the DB5.5 dataset consists of multi-chain proteins (these proteins are referred to as subassemblies), which can either be evaluated at the binary docking level (treating each subassembly as an individual entity) or multimeric docking (treating each chain as an individual entity).
>
> On the finetuning step for DB5.5, DiffDock-PP is finetuned at the level of binary docking, while DockGame is finetuned at the level of multimeric docking. We hypothesize that the additional improvement comes from the fact that the diffusion model in DockGame is trained to reason about relevant interactions to form subassemblies (by splitting them into constituent chains) while in DiffDock-PP, these subassemblies are left intact.
>
> **Experiments, Q3**: Thanks, we will incorporate this suggestion. However, a comparison solely based on best scores is not ideal for tasks like multimeric or binary docking, where typically multiple plausible structures exist, but only one is experimentally available.
>
> **Experiments, Q4**: Standard deviations are computed not only over the samples in the filtered set but also across all DB5.5 test complexes. This is also the case with the “best C-RMSD score” generations.
>
> **Experiments, Q5**: Thank you for the comment. We will clarify this in the paper, supported by an additional analysis of the TM-scores between different generations. The reviewer is correct that all the accepted samples should be from the same meta-stable state (indeed, this is our goal with such a filtering). However, even if structurally similar, such samples still exhibit some differences due to random noise injected at each diffusion step.

---

### Official Review · Reviewer_8gtr · 2023-10-31

**Soundness:** 2 fair
**Presentation:** 2 fair
**Contribution:** 2 fair
**Rating:** 3
**Confidence:** 4

**Summary:**

Given the common oversight in current methods, this paper introduces DockGame, a novel game-theoretic framework tailored for multimeric docking.

**Strengths:**

+ They are the first work to formulate the protein docking problem as a cooperative game.
+ They propose two approaches to compute plausible equilibria: (i) supervision from scoring functions; and (2) diffusion generative model.
+ Experimental results validate the effectiveness of the proposed model.

**Weaknesses:**

-	The motivation appears weak considering existing capabilities in multimeric docking and flexible models, such as Multimer, which have shown outstanding performance. The significance of designing an alternative rigid docking model remains unclear. Additionally, some generative models, like diffdock-pp and foldingdiff, seem straightforward to extend to multimeric docking.
-	There are some concerns about the contribution. The application of game theory does not introduce any additional benefits to the model design, as the derived diffusion framework and potential network would seemingly exist even without its inclusion.
-	There are confusions about the supervision of the energy function. Regarding the difficulty in learning the Boltzmann distribution, molecular simulation is both time-consuming and resource-intensive. I wonder how many samples are needed to learn potential functions. Besides, the inference process appears to mirror that of standard docking software, involving random sampling, evaluation, and ranking, which typically does not necessitate additional network training. Considering this, the necessity of the proposed potential network is unclear.
-	The separation of the two models within the paper suggests a fragmented approach. There is an inquiry into the possibility of utilizing energy to guide the learning of the diffusion generative model, potentially unifying the two models for a more cohesive structure.
-	Experimental results are not convincing enough. The limited size of the DB5.5 dataset raises concerns about the generalizability of the results to larger datasets such as DIPS. Furthermore, the absence of a comparison with HDock, a robust baseline for rigid docking, is notable. The paper also falls short in showcasing significant improvements compared to existing software baselines. The absence of essential ablation studies, such as those involving distance-based penalties, further undermines the strength of the experimental findings.

**Questions:**

Please refer to the weaknesses.

-	The motivation appears weak considering existing capabilities in multimeric docking and flexible models, such as Multimer, which have shown outstanding performance. The significance of designing an alternative rigid docking model remains unclear. Additionally, some generative models, like diffdock-pp and foldingdiff, seem straightforward to extend to multimeric docking.
-	There are some concerns about the contribution. The application of game theory does not introduce any additional benefits to the model design, as the derived diffusion framework and potential network would seemingly exist even without its inclusion.
-	There are confusions about the supervision of the energy function. Regarding the difficulty in learning the Boltzmann distribution, molecular simulation is both time-consuming and resource-intensive. I wonder how many samples are needed to learn potential functions. Besides, the inference process appears to mirror that of standard docking software, involving random sampling, evaluation, and ranking, which typically does not necessitate additional network training. Considering this, the necessity of the proposed potential network is unclear.
-	The separation of the two models within the paper suggests a fragmented approach. There is an inquiry into the possibility of utilizing energy to guide the learning of the diffusion generative model, potentially unifying the two models for a more cohesive structure.
-	Experimental results are not convincing enough. The limited size of the DB5.5 dataset raises concerns about the generalizability of the results to larger datasets such as DIPS. Furthermore, the absence of a comparison with HDock, a robust baseline for rigid docking, is notable. The paper also falls short in showcasing significant improvements compared to existing software baselines. The absence of essential ablation studies, such as those involving distance-based penalties, further undermines the strength of the experimental findings.

---

> ### Author Response · Authors · 2023-11-22
> **Author Response - 1/2**
>
> Thank you for your comments. We first aim to address the first two weaknesses jointly.
>
> **Motivation of Game-Theoretic Approach**
>
> As we discuss in the general response, the traditional paradigm in multimeric docking is to do pairwise docking between pairs of proteins, and then apply a combinatorial assembly algorithm to generate the final assembly structure. In contrast, the game-theoretic approach explores the combinatorial docking space by using local gradients (local wrt each protein’s actions) of the payoffs for each protein. This has an effect on the runtimes, where a traditional multimeric docking method like MultiLZerd takes about 20h to generate an assembly structure, while DockGame is able to generate a single structure in 4s (GPU vs CPU not withstanding).
>
> The framework can be extended to flexible multimeric docking by defining the protein specific payoff to be the common payoff plus a protein specific penalty term based on structural deviation from the native structure. This additional penalty term would only contribute a marginal to modest increase in runtimes.
>
> AlphaFold-Multimer indeed has great performance on multimeric docking, but is well known to be slow, and the performance degrades as the number of protein chains increases. In contrast, cooperative games, in principle, are more amenable to scalability, and gradient-based equilibrium computation provides an efficient exploration of the combinatorial docking space. Furthermore, one can compute multiple equilibria (either using diffusion models or gradients of potential network) in cooperative games, while AlphaFold-Multimer can only produce a single assembly structure estimate.
>
> **Comparisons to DiffDock-PP & FoldingDiff**
>
> While methods such as DiffDock-PP and FoldingDiff can be extended to the multimeric docking problem, this is something we also acknowledge in the Related Work section of the paper as well - “The closest related work to ours are (Corso et al., 2023; Ketata et al., 2023), of which our diffusion generative model for the cooperative game can be interpreted as a multi-agent extension.”
>
> While we agree with the reviewer that game-theory does not provide any additional benefit to the model design (the architectures of the potential network and diffusion model), we wish to clarify that the introduction of game-theory for multimeric docking problem was **not to improve model design**, but provide a **different perspective for approaching the problem**, with advantages and motivation highlighted above.
>
> **Connections between Diffusion Models and Multi-Agent Modeling**
>
> The connections between sampling in diffusion models and equilibrium computation in (cooperative) games are interesting in their own right. In the multi-agent literature, connections between sampling and equilibrium computation have been explored largely in the context of mean-field games (where one has infinite agents), or games with a large but finite but large number of homogenous agents.
>
> In contrast, our diffusion model does not place an assumption of the types of players, and provides a general recipe to define diffusion models over the (continuous) action spaces of all players in the cooperative game, when one only has access to equilibria and not the underlying game structure. Even if naturally obvious, this provides a connection between diffusion generative models and multi-agent system modeling, allowing for either community to benefit from respective advances.

---

> ### Author Response · Authors · 2023-11-22
> **Author Response - 2/2**
>
> We address the remaining weaknesses/questions below:
>
>
> >There are confusions about the supervision of the energy function...
>
> We use 10 perturbed samples per example to learn the potential function and train our model on 10000 examples from the DIPS dataset. We wish to emphasize that the potential function for only roto-translation perturbations (considered in this work) would be different than the one allowing for structural changes, with the latter being much more difficult to learn.
>
> Regarding the mirroring of our inference (equilibrium computation) process with those adopted by standard docking software, we wish to clarify some differences:
>
> Traditional *binary* docking software utilize random sampling, evaluation and ranking to generate docked structures. Traditional *multimeric* docking software however utilize a pairwise docking procedure (which builds on traditional binary docking), followed by a combinatorial assembly algorithm. This makes traditional methods like PyRosetta or MultiLZerd very slow, in addition to offering little to no support for GPU.
>
> In contrast, our inference process only uses randomization at initialization, does not involve any ranking but instead gradient-based updates which are fully supported on the GPU. In particular, as opposed to combinatorial assembly algorithms, our game-theoretic approach explores the combinatorial docking space by using local gradients (local wrt each protein’s actions) of the payoff. As a result, DockGame achieves much faster run-times than traditional multimeric docking methods (Table 1).
>
> >The separation of the two models ...
>
> Thank you for the remark. We developed the two approaches independently but our goal was to eventually unify them using the potential network as a guidance.
>
> >Experimental results are not convincing enough...
>
> For this work, we stuck to DB5.5 dataset for evaluation because it is the more traditionally used binary docking dataset, but can also support multimeric docking as many proteins are made up of multiple chains. This allowed us to compare DockGame at the level of multimeric docking to other baselines at the level of binary docking, in addition to traditional multimeric docking software like MultiLZerd.
>
> We outperform MultiLZerd significantly both in terms of runtime and other metrics, and have better median C-RMSDs than all methods except ClusPro, while being two orders of magnitude faster. HDock was left out of comparison because it is well known that their scoring function was fine-tuned on the DB4 dataset, which shares many examples with the DB5 dataset and makes their predictions look overoptimistic. We did not use DIPS for evaluation since the dataset only has examples with two chains while our goal was to evaluate DockGame at the level of multimeric docking.

---

> > ### Comment · Reviewer_8gtr · 2023-11-23
> > **Reply to the authors' response**
> >
> > Thanks for the detailed and thoughtful response from the author, addressing most of my concerns.
> >
> > However, there are still two key issues that haven't convinced me. (i) The core contribution of game theory, emphasized in the title, does not make a difference. New perspectives should bring some practical changes, but game theory has not provided new insights at both the model and theoretical levels. (ii) The concern about the effectiveness validation. Relying solely on the small DB5.5 dataset raises suspicions of overfitting and causes the lack of comparison with state-of-the-art baselines such as Multimer and HDOCK.
> >
> > In summary, at this point, I am still inclined to maintain the original score.

---

### Official Review · Reviewer_vQeX · 2023-11-08

**Soundness:** 2 fair
**Presentation:** 1 poor
**Contribution:** 2 fair
**Rating:** 3
**Confidence:** 5

**Summary:**

This paper focuses on the multimeric docking problem and proposes a game-theoretic framework for docking entitled DOCKGAME. In particular, DOCKGAME view protein docking as a cooperative game between proteins, where the ﬁnal assembly structure(s) constitute stable equilibria w.r.t. the underlying game potential.

**Strengths:**

1. The research question is meaningful to a subset of scholars within the ICLR community.
2. The methodology is sound.

**Weaknesses:**

1. The paper mentions the use of game theory, but in my view, the authors initially just use game theory as a narrative tool. The method does not actually involve game theory, which was somewhat disappointing when I read the methods section.
2. The presentation is not satisfactory.

**Questions:**

1. In the experimental section, why are there inconsistent assemblies provided by DOCKGAME-E and DOCKGAME-SE (one being 20 and the other 40)? This seems a bit unfair to DOCKGAME-E.
2. I would like to see the results of DOCKGAME on binary docking in isolation.
3. Can the proposed method be extended to torsional space?
4. I would like to see the results on the DockQ metric.

If my concerns are addressed effectively during the rebuttal period, I would be willing to revise my score.

---

> ### Author Response · Authors · 2023-11-22
>
> >The paper mentions the use of game theory, but in my view, the authors initially just use game theory as a narrative tool. The method does not actually involve game theory, which was somewhat disappointing when I read the methods section.
>
> We acknowledge your point of view, but respectfully disagree in this regard. We attempt to clarify this next.
>
> First, as we discuss in the general response, game theory is not just used as a narrative tool, but the introduced game-theoretic viewpoint is what allows us to view proteins as independent and potentially self-interested entities (in terms of deviations from their own native structures) with individual action spaces.
>
> This is in stark contrast with previous methods, which first apply a pairwise docking and then combinatorial assembly, and allows us to: 1) in principle, to scale efficiently to a large number of proteins and 2) naturally compute multiple plausible equilibria. See general response for a more detailed discussion about such points.
>
> In terms of methods, both our approaches are tightly bound to game theory. Indeed, in both cases proteins jointly update their strategies based on their local gradients (either coming from the surrogate potential function, or from the score network), which is a commonly used decentralized scheme in multi-agent learning to compute game equilibria. We acknowledge that this can also be viewed as local optimization of the potential function (because we consider a simple potential game), but the two do not coincide in case of more general payoff structures, e.g. when modeling flexible structures.
>
> >The presentation is not satisfactory.
>
> We appreciate the suggestion, and are happy to make relevant changes in this regard. Could you point to specific parts in the manuscript that would benefit from a change in presentation?
>
> >In the experimental section, why are there inconsistent assemblies provided by DOCKGAME-E and DOCKGAME-SE (one being 20 and the other 40)? This seems a bit unfair to DOCKGAME-E.
>
> This was purely due to our computational resources being limited when running such experiments. For DockGame-E, our goal was also to compute PyRosetta scores at each game iteration (to generate, e.g. Figure 2), and this required larger runtimes. We are now predicting 40 complexes for both methods in our current experiments.
>
> >Can the proposed method be extended to torsional space?
>
> Yes. The proposed method can be extended to the torsional space. However proteins are large graphs, and the modifications of torsional angles can have larger lever-arm effects on the protein structures, which can make training difficult.
>
> >I would like to see the results on the DockQ metric.
>
> DockQ metric has been developed in the context of binary docking. While the DockQ metric can be extended to multimeric docking by considering all pairs of interfaces in the assembly (and averaging them accordingly), this would imply a different set of comparisons between binary and multimeric docking methods. Our goal with the experiments was to highlight that DockGame can achieve comparable performance to binary docking methods while solving the multimeric docking task, and DockQ metrics would not have assisted in this regard.

---

> > ### Comment · Reviewer_vQeX · 2023-11-23
> > **Reply to the authors' response**
> >
> > Thank you for the authors' responses, which have effectively addressed some of my concerns.
> >
> > However, I have not received a satisfactory answer to my question regarding the connection between game theory and the methodology, and your reasons have not convinced me. Additionally, I have noticed that the other two reviewers also have this concern.
> >
> > Therefore, I have decided to maintain my original score for now.

---

### Official Review · Reviewer_4vSX · 2023-11-13

**Soundness:** 3 good
**Presentation:** 3 good
**Contribution:** 3 good
**Rating:** 5
**Confidence:** 5

**Summary:**

This paper models multimeric protein docking as a cooperative game. Specifically, each protein is a game member and they need to cooperate for multimeric docking. The reward is minimize some potential functions. Because the physical potential function is typically expensive to compute, they use two approaches to estimate true potential. one is learning a surrogate game potential from data generated by pyrosetta, the another is learning from data via diffusion generative model. The overall idea is interesting. i like this paper. However, the performance of this method is worse than baselines. This limitation makes this paper not ready to publish currently.

**Strengths:**

- modelling multimeric protein docking as a cooperative game is interesting.

- learning potential function from physical method, e.g., pyrosetta, is straightforward.

- learning potential function via diffusion model is not new but reasonable.

- this paper is well organized and well written.

**Weaknesses:**

- the major limitation of this paper is experiments. for example, the author filter results by TM-score and C-RMSD. What's performance without any filtering or ranking by model's potential.

- It's interesting to show the distribution of decoy quality and the sampling efficient.

- although alphafold-multimer already used data in test, i think it's necessary to curate a new small data set from latest PDB. alphafold-multimer is a strong baseline.

- in the section of "Equilibrium Computation via Gradient-Based Learning", filtering decoys by Euclidean distance only could not be sufficient. i am not sure how noisy is the training data if you only use this filter. i guess this heuristic approach could limit the performance of DOCKGAME-E (DOCKGAME-E is worse than DOCKGAME-SM from table 1)


missing some key references:

[1] Bryant, Patrick, et.al. Predicting the structure of large protein complexes using AlphaFold and Monte Carlo tree search

[2] Yujie Luo, et.al. xTrimoDock: Rigid Protein Docking via Cross-Modal Representation Learning and Spectral Algorithm.

[3] Ambrosetti, Francesco, et.al. Modeling antibody-antigen complexes by information-driven docking.

[4] Ghani, Usman, et.al. Improved docking of protein models by a combination of alphafold2 and cluspro

**Questions:**

- is this method robust on single's structure? for example, if you use predicted protein structure rather unbound native structure, what's the performance.

---

> ### Author Response · Authors · 2023-11-22
>
> >the major limitation of this paper is experiments. for example, the author filter results by TM-score and C-RMSD. What's performance without any filtering or ranking by model's potential.
>
> The reason we operate filtering by TM score, is that DockGame naturally finds multiple plausible types of docked versions (plausible equilibria), while the test data consists of a single docked complex. Thus, to cope with this, we select the subset of generated complexes that are structurally similar to the bound version (as measured by TM score). We will clarify this point further and will add complete metrics distributions.
>
> >It's interesting to show the distribution of decoy quality and the sampling efficient.
>
> These are both key features of DockGame. Indeed, compared to previous work, DockGame 1) can compute multiple plausible equilibria for each protein and 2) scales favorably with the number of chains. We acknowledge that both these messages were not delivered sufficiently well in our experiments. We will add two additional plots and discussions illustrating these points.
>
> > although alphafold-multimer already used data in test, i think it's necessary to curate a new small data set from latest PDB. alphafold-multimer is a strong baseline.
>
> Thank you for the suggestion. To compare to AlphaFold-Multimer, we would indeed need to curate a different test set from PDB. This would also imply curating an appropriate large training set and retraining our methods.
>
> >in the section of "Equilibrium Computation via Gradient-Based Learning", filtering decoys by Euclidean distance only could not be sufficient. i am not sure how noisy is the training data if you only use this filter. i guess this heuristic approach could limit the performance of DOCKGAME-E (DOCKGAME-E is worse than DOCKGAME-SM from table 1)
>
> The Euclidean distance constraint is *only* an inference-time addition and is not applied to the decoys generated at training time. The decoys are generated at training time for the purpose of learning an (approximate) potential function.
>
> Furthermore, as mentioned in the section, this constraint is added to prevent proteins in the assembly from being far away in the 3D space (this would correspond to protein undocking), which are also equilibria wrt the potential. The distance penalty also contributes to the gradient-based equilibrium computation only if the distances between proteins is greater than the specified threshold.
>
> The evaluation in Table 1 measures C-RMSD, which will only be lower if the predicted assembly is close in an RMSD sense to the true assembly. Thus the addition of the distance constraint does not limit the performance of the DockGame-SM, since its only function is to discourage proteins to be too far away from each other.
>
> >missing some key references:...
>
> Thank you for the pointers. Reference [1] is already in the paper. We will add the remaining references.
>
> >is this method robust on single's structure? for example, if you use predicted protein structure rather unbound native structure, what's the performance.
>
> Currently, DockGame only uses rotations and translations as the action space for each protein. This means we start with the true assembly structure and apply roto-translation perturbations to each protein to learn the underlying game potential. We could indeed start with predicted structures and compute the assembly structure, but the predictions would be way off for examples where the native structure of a protein undergoes larger structural changes in the assembly.
>
> The current framework would have to be extended to include structure-based penalty terms in the equilibrium computation step to improve the model’s robustness, which we believe is beyond the current technical scope of the paper, and is left to future work.

---

### Author Response · Authors · 2023-11-22
**General response**

We thank all the Reviewers for their constructive comments and questions regarding our work.
While we address their specific questions individually, we provide here a general response about the novelty of our work and the benefits of our game-theoretic modeling. We acknowledge this might have not come across clearly from the paper and we hope this clarifies some of the raised concerns.

At a high level, our game-theoretic viewpoint on multimeric docking brings the main novelty of considering each protein as an independent and potentially self-interested player (in terms of deviation from its own native structure) in a game. This is in stark contrast with traditional approaches to multimeric docking that first apply pairwise docking between pairs of proteins, and then utilize a combinatorial assembly algorithm to generate the final assembly structure.

In particular, our game-theoretic viewpoint introduces the following benefits:

- The combinatorial space of all possible docking configurations is broken down into individual agents’ actions which are updated by only using local gradients (local wrt each protein’s actions) of the payoffs. **The game perspective allows us to model multimeric docking as a gradient-based optimization problem instead of the traditionally used combinatorial assembly paradigm**. This has a large effect, e.g., on runtimes, where a traditional multimeric docking method like MultiLZerd takes about 20h (only on CPU) to generate an assembly structure, while DockGame is able to generate a single structure in 4s (by leveraging GPUs).

- DockGame naturally allows one to compute multiple and diverse docking structures (which are viewed as plausible game equilibria), as opposed to trying to regress towards a single docked version as in previous approaches. This fact was largely neglected by previous works where, indeed, complexes are evaluated via the C-RMSD from the *only* equilibrium present in the dataset. We believe additional metrics such as the diversity of equilibria and their quality should be considered to cope with this fact.

- Although we consider proteins with aligned interests (specified in terms of a common potential), our framework can naturally be extended to protein-specific payoffs to model, e.g., *flexible* multimeric docking. In such a case, each player’s payoff can be represented by a common payoff plus a protein-specific penalty term based on structural deviation from the native structure. This additional penalty term would only contribute a marginal to a modest increase in runtimes.

---

### Meta-Review · Area_Chair_Qavo · 2023-12-05

**Metareview:**

This paper addresses the problem of multiple protein docking using a cooperative game theoretic approach. The reviewers raised several concerns during the review process. It was unclear why the authors chose to use a surrogate potential function when potential functions exist in software such as Rosetta. Second, the formulation of the loss function does not seem to align with the gradient of the energy function. The authors responded reasonably to these concerns that there is limited support in Rosetta for GPU computations and the loss function does, indeed, align with the energy function gradients. Overall, the authors made reasonable responses to the concerns of the reviewers, and changes to the paper will improve the clarity. But the amount of revision needed would be substantial and perhaps outside of the scope of review process.

**Justification For Why Not Higher Score:**

The reviewers raised many valid concerns. If addressed, the paper would be much improved, but the revisions are substantial.

**Justification For Why Not Lower Score:**

N/A

---

### Decision · Program_Chairs · 2024-01-16

Reject